# Management of Large Subcardial Diverticula in Sleeve Gastrectomy: Technical Tips

**Francesco Frattini** [1,*], **Antonella Pino** [1], **Giuseppe Cordaro** [1], **Georgios Lianos** [2], **Simona Bertoli** [3] **and Gianlorenzo Dionigi** [1]

1    Division of Surgery, IRCCS Istituto Auxologico Italiano, 20122 Milan, Italy
2    Department of Surgery, University of Ioannina, 45332 Ioannina, Greece
3    Obesity Unit, IRCCS Istituto Auxologico Italiano, University of Milan, 20122 Milan, Italy
*    Correspondence: f.frattini@auxologico.it

**Abstract:** Gastric diverticula are uncommon anatomic abnormalities that are usually asymptomatic or found incidentally in upper gastrointestinal radiographs with contrast or endoscopy. Gastric diverticula usually arise from the wall of the gastric fundus. Their preoperative study or intra-operative finding is of great importance in patients with obesity who are scheduled for bariatric surgery. In bariatric surgery, and especially in sleeve gastrectomy, it is of utmost importance to know the exact location of the diverticulum in order to position the stapler correctly and to perform appropriate gastric resection including the diverticulum. Sleeve gastrectomy has gained popularity worldwide and currently accounts for the most performed bariatric procedure according to more recent international surveys. It is considered to be a technically easy procedure. Nonetheless, some steps of the procedure, such as gastric fundus mobilization and the gastric resection with the use of the stapler, may be challenging in patients with a high BMI and in the presence of abnormalities of the gastric wall. This can represent a risk for the occurrence of complications such as a gastric leak or bleeding. We propose some considerations about technical tips to adopt for safely performing sleeve gastrectomy in the presence of a subcardial diverticulum.

**Keywords:** bariatric surgery; sleeve gastrectomy; gastric diverticulum; esophagogastroduodenoscopy; staple-line





## 1. Introduction

Sleeve gastrectomy has become the most performed bariatric procedure worldwide. It is considered to be a safe and effective procedure in patients with obesity in terms of weight loss and the improvement of comorbidities and their quality of life. It is not technically challenging and has a lower complication rate in comparison to that of a gastric bypass. Nonetheless, increasingly large series and longer follow-ups have demonstrated troublesome complications other than gastric leaks and bleeding, such as new onset gastro-esophageal reflux and porto-mesenteric vein thrombosis. The standardized steps in this operation must be accurately respected because technical defects in dissection or stapling might be cause of insidious complications such as bleeding or leaks. This complication risk might be higher in the presence of gastric lesions such as diverticula, especially if they have been unexpectedly found during surgery and have not been preoperatively diagnosed.

The aim of this article is to propose some technical tips to adopt in performing sleeve gastrectomy in the presence of a subcardial diverticulum.

### 1.1. Gastric Diverticula and Sleeve Gastrectomy

Gastric diverticula are a rare entity with incidences of 0.12% in computed tomography scans and of 0.01–0.10% in endoscopy [1–3]. They can be congenital or acquired and are commonly asymptomatic, usually incidentally found during upper gastrointestinal endoscopy or upper gastrointestinal contrast radiographic studies [4].

Nonetheless, gastric diverticula can be a cause of complications such as a hemorrhage, perforation, abscess, or malignancy [5–8].

The posterior wall of the upper gastric fundus is the most frequent site of occurrence.

In the upper gastrointestinal series, a gastric diverticulum may show up as an image of saccular pouch of the gastric wall at the hydro-air level (shown in Figure 1).

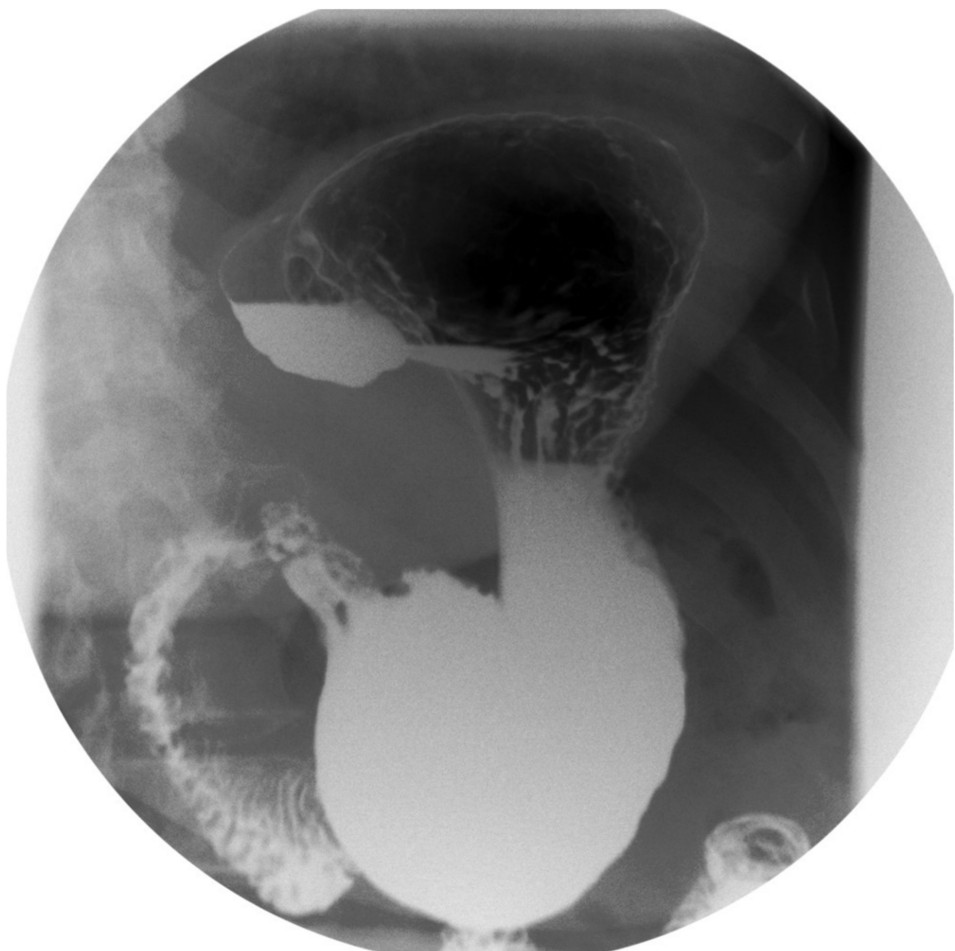

**Figure 1.** Preoperative upper gastrointestinal series showing diverticulum of the upper gastric fundus.

In the literature, some clinical cases of gastric diverticula have been reported, but there is little evidence on their management in bariatric surgery [9,10].

A literature search was conducted in PubMed, EMBASE, and Web of Science databases for the terms "bariatric surgery", "sleeve gastrectomy", and "gastric diverticulum".

In the two cases presented by Castelli et al. [9], the gastric diverticulum was incidentally found during greater curvature dissection in one case, whereas it was endoscopically diagnosed before bariatric surgery in the other case. In one case, the diverticulum spontaneously perforated during the dissection. Spontaneous perforation of the diverticulum shows that the wall of the diverticulum is delicate and fragile. Manipulation and dissection of the gastric diverticulum must be accurate and careful to prevent perforation, which may contaminate the field and lengthen the surgical time.

In the report by Wylie et al. [10], the resection of the diverticulum was completed concurrently with a sleeve gastrectomy in a patient with obesity, symptomatic of the diverticulum. The preoperative endoscopy and upper gastrointestinal series revealed the diverticulum on the posterior gastric wall.

Even if sleeve gastrectomy has been considered as a relatively easy procedure, the complete dissection of the gastric fundus and the gastric resection have to be meticulously and accurately performed to prevent complications such as gastric leaks and bleeding.

There have been significant controversies regarding several preoperative, perioperative, and postoperative issues involving sleeve gastrectomy. Some consensus surveys [11] and the recent Delphy Consensus [12] from a multinational team of experts have provided statements on these controversial issues.

One of the statements of the Delphy Consensus reported that patients should undergo a mandatory, routine preoperative upper gastro-intestinal endoscopy (Agree 79.2%). This is a conditional recommendation also in the EAES guidelines for bariatric surgery update from 2020.

The presence of gastric diverticula, such as other gastric lesions as stromal tumors, neuroendocrine tumors, malignant tumors, ulcers, or polyps, that have not been diagnosed preoperatively at the esophagogastroduodenoscopy could represent a serious concern to a safe dissection and stapling resection.

### 1.2. Technical Considerations

As assessed by the different editions of the Sleeve gastrectomy Consensus Conference and by the Delphy Consensus, the steps of sleeve gastrectomy are the following:

—   Opening a window in the gastro-colic ligament using an energy device from the antrum and proceeding up to the angle of His, along the greater curvature, until the left diaphragmatic pillar was fully exposed.
—   The gastric fundus must be entirely mobilized with coagulation and section of the short gastric vessels and the opening of the gastro-splenic ligament.
—   After the transoral position of a bougie (of diameter ranging 36–40 Fr), sleeve gastrectomy is performed with the use of a stapler, beginning from the distance of 4–6 cm from the pylorus up to the angle of His.
—   Care must be taken in the first two uses of the stapler to prevent narrowing the gastric sleeve at the incisura.
—   Symmetric traction on the anterior and posterior stomach wall must be undertaken during stapling.
—   The staple line should be regular and straight from the antrum to the angle of His.
—   Each reload of the stapler must be calibrated based on the thickness of the gastric wall, as well documented in the work of Elariny et al. [13], providing the right compression and hemostasis of the tissue.
—   The staple line should stay at least 1 cm away from the esophago-gastric junction.

In presence of a gastric diverticulum, the dissection of the upper gastric fundus on the posterior wall might be a challenging step. As shown in Figures 2 and 3 of a case we treated, a large diverticulum of the posterior gastric fundus with a wide-based implantation was identified during the greater gastric curvature dissection, after division of the short gastric vessels (shown in Figures 2 and 3). The diverticulum was 2 cm away from the esophago-gastric junction and its neck and very close to the posterior of the pancreas and to the splenic artery course just before its branching at the splenic hilum. The diverticulum showed a frail wall that is thinner than the normal gastric wall in the dissection maneuver (Figures 4 and 5).

In the guide, a 36 Fr oro-gastric bougie sleeve gastrectomy was completed using a linear flexible stapler with green and gold reloads. The postoperative course was uneventful, and the patient was discharged three days after the surgery. The upper gastrointestinal study on the second postoperative day did not reveal any leakage of gastrografin from the staple line (Figure 6).

In patients scheduled for bariatric surgery, preoperative diagnosis with esophagogastroduodenoscopy and/or the upper gastrointestinal series allows practitioners to perform sleeve gastrectomy safely and to know the site of the diverticulum and its distance from the esophago-gastric junction. Some troubled issues must be considered: the site of the

diverticulum and its proximity to the lesser or the greater curvatures, the distance of the neck of the diverticulum from the esophago-gastric junction, and its posterior extension up to the upper margin of the pancreas.

Gastric diverticula are usually found during sleeve gastrectomy after greater curvature dissection and sharp division of the retrogastric and pre-pancreatic attachments.

During placement of the stapler for sleeve gastrectomy, special attention must be taken when one is pulling the diverticulum laterally from the staple line and completely including it within the resected specimen (Figures 2 and 3). Special attention must be paid to avoid including the diverticulum or its neck into the staple line. The wall of the diverticulum could be a point of weakness that is predisposed to gastric leaks.

Gastric diverticula of the posterior wall of the upper gastric fundus are usually close to the anterior surface of the pancreas, on the superior margin, and to the splenic artery course. Therefore, their dissection must be careful to avoid intraoperative complications such as splenic lesions, pancreatic lesions, or splenic artery lesions.

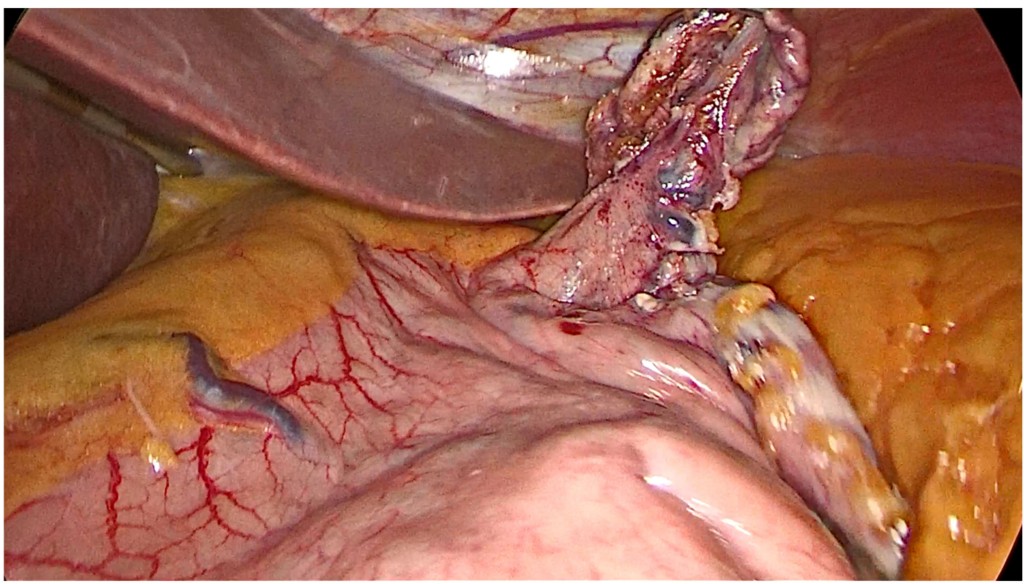

**Figure 2.** Intraoperative image of the gastric diverticulum after dissection of the greater curvature.

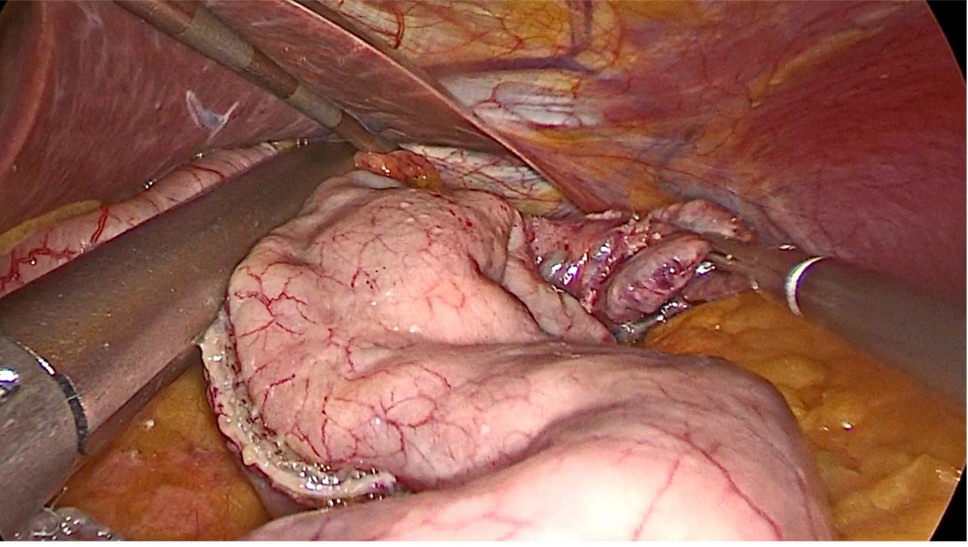

**Figure 3.** Intraoperative image of sleeve gastrectomy with concomitant resection of the diverticulum.

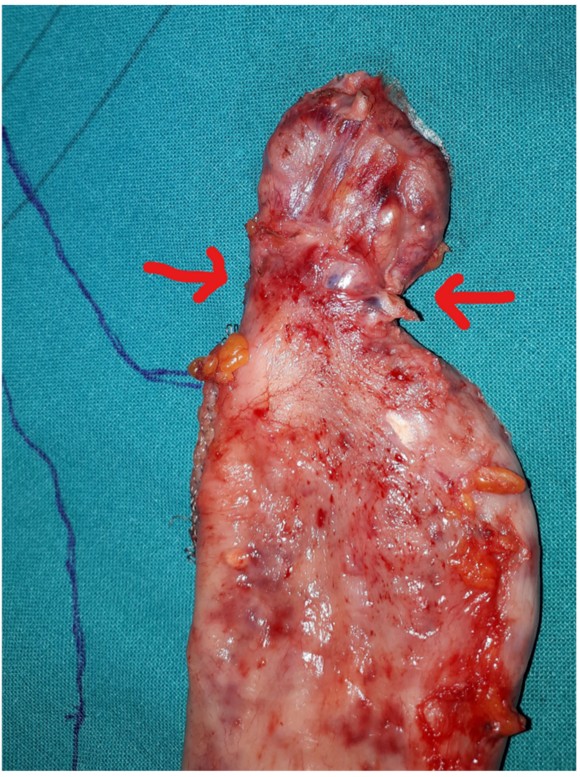

**Figure 4.** Gastric specimen after sleeve gastrectomy. Red arrows: Neck of the diverticulum.

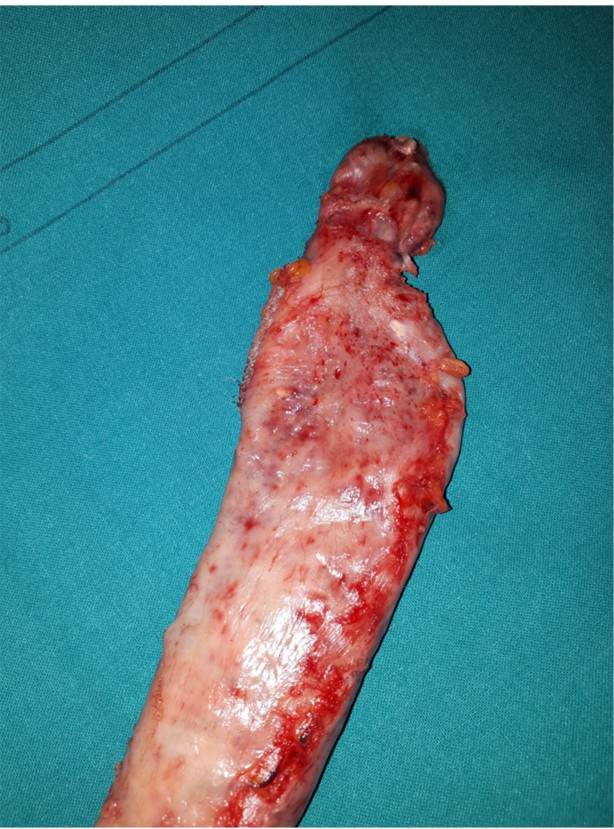

**Figure 5.** Gastric specimen with large diverticulum of the upper fundus.

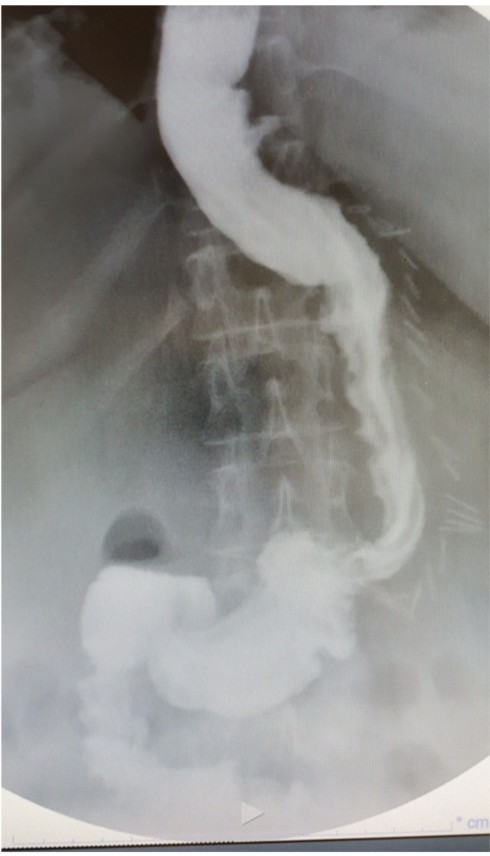

**Figure 6.** Postoperative upper gastrointestinal series.

After an incidental finding of gastric diverticulum that was not diagnosed in the preoperative endoscopic or imaging work-up, it became necessary to accurately evaluate the resectability of the diverticulum in the sleeve gastrectomy.

If the diverticulum cannot be simultaneously excised in the sleeve gastrectomy because of its size or its position being too close to the esophago-gastric junction or to the lesser curvature, another surgical strategy will have to be evaluated. If sleeve gastrectomy cannot completely remove the diverticulum, it is mandatory to consider what is the best option to follow. If the diverticulum is asymptomatic, is it necessary to remove it before second-step sleeve gastrectomy?

## 2. Conclusions

The preoperative diagnosis of a gastric diverticulum by endoscopy or an upper gastrointestinal swallow study in a patient scheduled for bariatric surgery is of great importance to plan a safe and correct surgical procedure.

Complete dissection of the diverticulum must be careful and accurate. If the diverticulum is subcardial, great care must be taken in maintaining a safe distance from the esophago-gastric junction and keeping the diverticulum away from the staple line.

**Author Contributions:** F.F. wrote the main manuscript text, A.P. and G.C. prepared Figures 1–3. F.F., A.P., G.C., G.L., S.B. and G.D. revised the manuscript. All authors have read and agreed to the published version of the manuscript.

**Funding:** This research received no external funding.

**Institutional Review Board Statement:** Not applicable.

**Informed Consent Statement:** Written consent has been obtained from the patient to publish this paper.

**Data Availability Statement:** Not applicable.

**Conflicts of Interest:** All the authors have no competing interests of financial or personal nature.

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
