# Peer review of "Management of Large Subcardial Diverticula in Sleeve Gastrectomy: Technical Tips"

_2673-4095, doi:10.3390/surgeries4010015_

Round 1

Reviewer 1 Report

Please correct "Aim of this article" to "The aim of this article"

Please correct "in presence of a subcardial diverticulum" to "in the presence of a subcardial diverticulum"

Please do not use contractions such as "didn’t"

While this is a short and interesting paper, the breadth of its suggestions revolve only around the management of a single case. It would be more interesting if the authors could provide alternate techniques based on more experiences from surgeons within the research team or elsewhere. Otherwise, this  reads more like a case report.

It would also be helpful if the authors could create a diagram illustrating the technical steps to this operation.

Author Response

In the revised paper the corrections indicated by the reviewer have been added.

The technical steps of the operation have been accurately reported and described.

Reviewer 2 Report

This is a "honest" paper that remarks the need for a proper and accurate dissection of gastric fundus to carry out a well-designed and shaped LSG.

Some text is missing in line 32 "accurately ... because" and must be amended

Author Response

The missing test in line 32 has been added as suggested

Reviewer 3 Report

The authors have present a case with gastric diverticulum underwent LSG. Technical article requires intraoperative pictures for emphasizing their experience.

1. Please provide some intraoperative pictures for emphasizing your experience.

2. Please show the postoperative enema.

3. Gastric diverticulum is often located at posteiror wall of the gastric  fornix. Please show the previous reported cases underwent LSG with gastric diverticulum. 

Author Response

As suggested by the reviewer I provided the picture of the postoperative UGI series and the intraoperative picture of the diverticulum after dissection.

I reported also other cases described in literature about gastric diverticulum treatment in sleeve gastrectomy.

Round 2

Reviewer 1 Report

Thank you to the authors for addressing my comments.

Reviewer 3 Report

The authors have described their manuscript along with our suggestion, and this paper has become more informative for readers.